# Antitubercular activity assessment of fluorinated chalcones, 2-aminopyridine-3-carbonitrile and 2-amino-4H-pyran-3-carbonitrile derivatives: *In vitro*, molecular docking and in-silico drug likeliness studies

**Surendra Babu Lagu**[1]*, **Rajendra Prasad Yejella**[2], **Srinath Nissankararao**[3], **Richie R. Bhandare**[4,5]*, **Venu Sampath Golla**[2], **Bontha Venkata Subrahmanya Lokesh**[6], **M. Mukhlesur Rahman**[7], **Afzal Basha Shaik**[8]*

1 Pharmaceutical Chemistry Division, Adikavi Nannaya University College of Pharmaceutical Sciences, Adikavi Nannaya University, Tadepalligudem, Andhra Pradesh, India, 2 Department of Pharmaceutical Chemistry, University College of Pharmaceutical Sciences, Andhra University, Visakhapatnam, Andhra Pradesh, India, 3 Montvale, New Jersey, New Jersey, United States of America, 4 Department of Pharmaceutical Sciences, College of Pharmacy & Health Sciences, Ajman University, Ajman, United Arab Emirates, 5 Center of Medical and Bio-allied Health Sciences Research, Ajman University, Ajman, United Arab Emirates, 6 Department of Pharmaceutical Chemistry, Faculty of Pharmacy, University of Malaya, Kuala Lumpur, Malaysia, 7 Medicines Research Group, School of Health, Sports and Bioscience, University of East London, London, United Kingdom, 8 Department of Pharmaceutical Chemistry, Vignan Pharmacy College, Jawaharlal Nehru Technological University, Vadlamudi, Andhra Pradesh, India

* bashafoye@gmail.com (ABS); r.bhandareh@ajman.ac.ae (RRB); ysbabu033@gmail.com (SBL)

## Abstract

A series of newer previously synthesized fluorinated chalcones and their 2-amino-pyridine-3-carbonitrile and 2-amino-4H-pyran-3-carbonitrile derivatives were screened for their *in vitro* antitubercular activity and *in silico* methods. Compound **40** (MIC~ 8 µM) was the most potent among all 60 compounds, whose potency is comparable with broad spectrum antibiotics like ciprofloxacin and streptomycin and three times more potent than pyrazinamide. Additionally, compound **40** was also less selective and hence non-toxic towards the human live cell lines-LO2 in its MTT assay. Compounds **30**, **27**, **50**, **41**, **51**, and **60** have exhibited streptomycin like activity (MIC~16–18 µM). Fluorinated chalcones, pyridine and pyran derivatives were found to occupy prime position in thymidylate kinase enzymatic pockets in molecular docking studies. The molecule **40** being most potent had shown a binding energy of -9.67 Kcal/mol, while docking against thymidylate kinase, which was compared with its *in vitro* MIC value (~8 µM). These findings suggest that 2-aminopyridine-3-carbonitrile and 2-amino-4H-pyran-3-carbonitrile derivatives are prospective lead molecules for the development of novel antitubercular drugs.

**Data Availability Statement:** All relevant data are within the paper.

**Funding:** Our manuscript study did not receive any external funding, however partial funding towards article processing charges has been provided by Deanship of Graduate Studies and Research, Ajman University and the same has been included in the manuscript. The funders had no role in study design, data collection and analysis, decision to publish, or preparation of the manuscript.

**Competing interests:** The authors have declared that no competing interests exist.

## Introduction

TB is responsible for the death of 1.4 million patients and among them 14.86% (2,08,000) were associated with HIV. *Mtb* is an agile bacterium which can shift from active to latent stage and vice-versa. Additionally, Tuberculosis (TB) is caused by the bacteria *Mycobacterium tuberculosis* (*Mtb*), which has been a global concern since decades and big burden on the healthcare system till today. It is one of the top ten leading causes of death according to the World Health Organization (WHO) 2019 reports and pose multiple resistance from the immunological protection of the host [https://www.who.int/news-room/fact-sheets/detail/the-top-10-causes-of-death]. These typical microbiological features of *Mtb* exhibit for long term therapy and even multi drug resistance (MDR) to the existing WHO treatment protocols. However, due to improved diagnosis and treatment, the incidence of deaths from tuberculosis has been reduced in recent years. It's anticipated that 66 million were rescued from the death during the year 2000–2020 [https://www.who.int/news-room/fact-sheets/detail/tuberculosis]. Nevertheless, the increase in the incidence of MDR and extensively drug-resistant (XDR) *Mtb variants* may complicate and create new challenges to this progress. In recent reports comparison in 2018, the number of patients with MDR or rifampicin-resistant TB (RR-TB) has increased by 10% in 2019. India has been listed in the top countries list for RR-TB [1]. MDR-TB, XDR-TB and RR-TB are posing greater challenges in the current treatment tactics employed. This scenario made scientists to drive and emphasize the significance of new research findings and development of new antitubercular medicines to compliment and supplement the current treatment protocols [2].

Chalcones are open chain flavonoids and have received large attention due to their diverse biological activities and synthetic applications in the preparation of medicinally useful heterocyclic molecules [3, 4]. They possess valuable antimicrobial, anti-inflammatory, antitubercular and anticancer activities [5–9]. Among them, fluorinated chalcones (Fig 1) were reported with excellent antimycobacterial activities [10–12]. Pyridine and pyran are biologically useful heterocycles with excellent therapeutic utility. Both these scaffolds are distributed in many naturally derived compounds like alkaloids, flavonoids, glycosides etc. Pyridine derivatives were reported to possess sundry of activities like anticancer, antitubercular, antiviral, antimicrobial, anti-inflammatory, antihistaminic etc., [13–27] and this ring is also present in a large variety of drugs used for the treatment of different diseases [28]. Pyran derivatives have been reported to exhibit antibacterial, anticancer, antitubercular, antioxidant, and other properties [29–37]. Antitubercular medications such as isoniazid, ethionamide, and prothionamide use the pyridine scaffold, whereas aminoglycoside antitubercular medications such as streptomycin, kanamycin, and amikacin use the reduced version of the pyran ring, tetrahydropyran (Fig 1). In the recent past, different fluorinated chalcones, pyridine and pyran derivatives were identified as potential antitubercular lead compounds [38–40] (Fig 2).

Fluorine substituents would change the physicochemical properties of drug-like candidates and drugs, such as increased penetration through biological membranes and metabolic stability, lower clearance, pKa value, better binding characteristics and potency [41–44]. Due to its similar sizes, the pyridine ring is a typical bioisosteric scaffold for benzene, pyrrole, and oxazole. In addition, it is utilized to substitute amines and amides bioisosterically. Basicity, stability, reduced molecular size, and aromatic nature confer superior pharmacokinetic qualities and binding abilities to pyridine, making it suitable for use in the preparation of lead molecules [24, 45]. Pyran scaffold like many other heterocyclic rings, aid bioactive compounds in altering pharmacokinetic and pharmacodynamic properties that make it an important component of many pharmaceuticals [46, 47]. In our previous published studies, we reported the antibacterial and antifungal properties of fluorinated chalcones and their pyridine and pyran derivatives

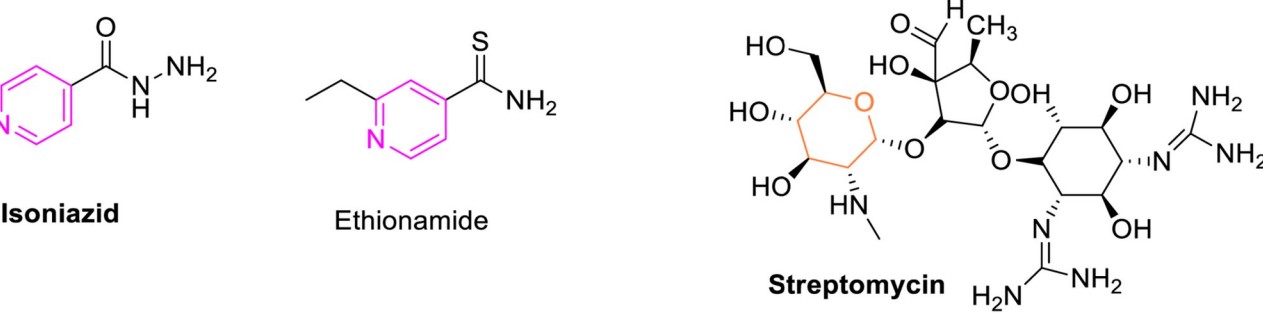

**Fig 1. Structures of selected marketed antitubercular drugs containing pyridine and tetrahydropyran scaffolds.**

with promising activities [48, 49]. Motivated by the excellent antimicrobial activities, we intended to test the target compounds against *Mycobacterium tuberculosis* in order to identify potential lead molecules against tuberculosis infections. Hence, herein we report the antitubercular evaluation of fluorinated chalcones and their 2-amino-pyridine-3-carbonitrile and 2-amino-4H-pyran-3-carbonitrile derivatives by standard computational and biological methods.

## Materials and methods

### *In vitro* antitubercular study

The antimycobacterial activity of target compounds (**1–60**) were evaluated against *Mycobacterium tuberculosis* using MABA. Briefly, sterile deionized water (200 μL) was added to all outer perimeter wells of sterile 96 well plate to minimize evaporation of medium in the test wells during incubation. 100 μL of the Middlebrook 7H9 broth was further added and serial dilution

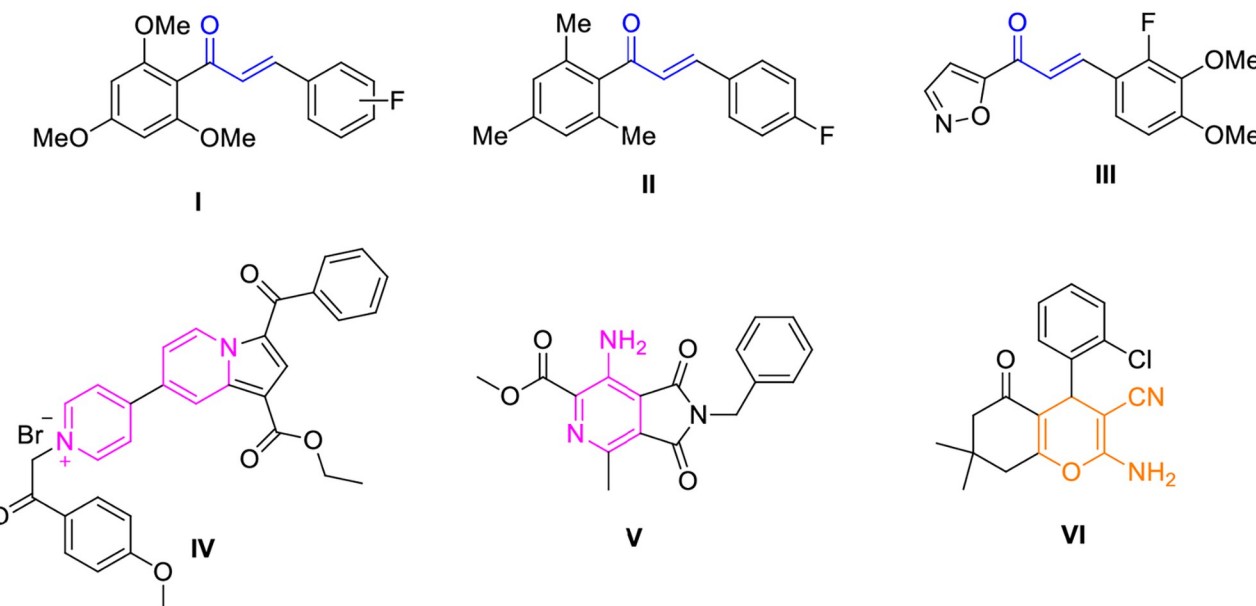

**Fig 2. Fluorinated chalcones, pyridine and pyran derivatives with significant antitubercular activity.**

of compounds were made directly on plate. The final drug concentrations obtained were 100 to 0.2 μg/mL. The plates were later covered and sealed with parafilm and incubated at 37˚C for five days. After incubation, 25 μL of freshly prepared 1:1 mixture of Alamar Blue reagent and 10% tween 80 was added to the plate and incubated for 24 hrs. A blue color in the well was interpreted as no bacterial growth, and pink color was scored as growth. The MIC was defined as lowest drug concentration which prevented the color change from blue to pink [50, 51]. Additionally, cLogP of all the target compounds has been calculated using SwissADME software.

### Molecular docking studies

The molecular docking studies on the crystallographic structure of Thymidylate Kinase (PDB ID: 1G3U) retrieved in protein data bank was performed using AUTODOCK 4.2 version [52]. The AUTODOCK TOOLS were utilized for preparing the protein for docking. The polar hydrogen's, partial charges and Gastegier charges were added using these tools. The flexible torsion of the ligand was assigned to proteins; the Auto Grid tool was used to implement the auto grid file, different for each protein. The grid parameters were set for each run. The Auto-Dock was run for binding molecules for molecular docking into the crystallographic structure of 1G3U active site. The cube selected covers the protein only of the binding site. The docking log file for each run was used to generate top three binding affinities of each protein, based on the binding energy of compounds. The saved 3D molecular poses were visualized using Discovery Studio.

### MTT assay

The most active antitubercular lead compound **40** was screened for its cytotoxicity by MTT assay against the human normal liver cell lines (LO2) to assess its selectivity and safety using the procedure described as per literature [48].

### In silico drug likeliness prediction

SwissADME, a web tool was used to evaluate the properties of the most potent compounds **20**, **37**, **40**, and **60**, as well as the marketed antitubercular drugs Ciprofloxacin, Streptomycin, and Pyrazinamide, for their in-silico parameters such as GI absorption, Lipinski rule of five, and CYP2C19 and CYP2D6 inhibition, in order to meet the requirements of the drug-likeliness. (http://www.swissadme.ch/ (accessed on 28 June, 2021)) [53].

## Results and discussion

### Chemistry

The synthesis and characterization of compounds **1–60** has been published previously [48, 49]. Fig 3 depicts the synthesis of target compounds (**1–60**).

### In vitro antitubercular activity

Synthesized compounds (**1–60**) were evaluated for their antitubercular activity against tubercular strain H37R$_V$ (**Tables 1–3**, refer to S1 File). As reference standards, ciprofloxacin (MIC~9 μM), pyrazinamide (MIC ~25 μM), and streptomycin (MIC~11 μM) were utilized. The synthesized compounds can be classified as trifluoromethyl/trifluoromethoxy containing chalcones (**1–20**); 2-amino-pyridine-3-carbonitrile (**21–40**) and 2-amino-4H-pyran-3-carbonitrile (**41–60**) derivatives. The phenyl ring (R"; **Tables 1–3**, refer to S1 File) was substituted with electron withdrawing groups at ortho, meta, para or ortho-meta positions (compounds

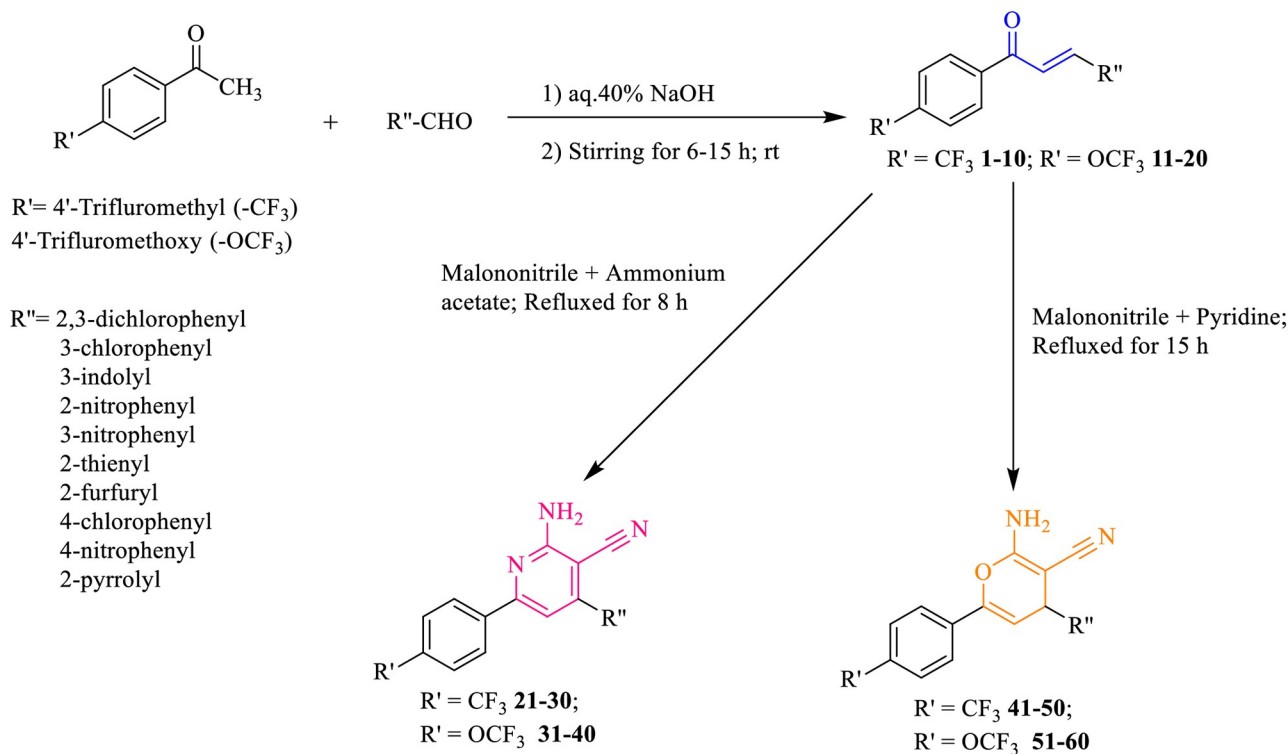

**Fig 3. Scheme 1, Synthesis of fluorinated chalcones (1–20) and their 2-amino-pyridine-3-carbonitrile (21–40) and 2-amino-4H-pyran-3-carbonitrile (41–60) derivatives.**

**1–6**, **11–16**, **21–26**, **31–36**, **41–46**, and **51–56**). Various heterocycles such as thiophene, furan, pyrrole, and indole ring systems were considered to examine bioisosteric substitution of the phenyl ring (compounds **7–10**, **17–20**, **27–30**, **37–40**, **47–50** and **57–60**). In case of trifluoromethyl and trifluoromethoxy chalcone series (**1–20**), the MICs ranged from 38–189 μM (**Table 1**, refer to S1 File). In trifluoromethyl substituted chalcone derivatives, substituting Cl (**10**) in the para position resulted in a MIC of 40 μM, whereas substituting Cl in the meta (**2**)/ ortho-meta (**3**) position or using nitro group in the ortho (**4**), meta (**5**), or para (**6**) position did not improve antitubercular activity when compared to the reference standards. Compounds **10** and **20** had the best activity among the bioisosteres, with MIC values of 40 and 38 μM, respectively.

In case of trifluoromethyl and trifluoromethoxy substituted 2-amino-pyridine-3-carbonitrile series (**21–40**), the range of MICs values were obtained from 8–134 μM. Substituting electronegative group (Cl or $NO_2$) at positions *ortho* or *para* was found to be favorable for activity (compounds **21**, **31**, **23**, **33**, **24**, **34**). However, the compounds bearing heteroaryl scaffold (**27– 30**, **37–40**) were more active than the compounds containing phenyl ring with electronwithdrawing groups. Among them, compound **40** containing indole scaffold was most potent with a better MIC value (8 μM). This compound **40** was found to have activity similar to ciprofloxacin and streptomycin but was 3-fold potent than pyrazinamide.

Wheras the trifluoromethyl and trifluoromethoxy substituted 2-amino-4H-pyran-3-carbonitrile series (**41–60**) were shown the variation of MIC values (16–161 μM). Substitution of the electro negative group (Cl or $NO_2$) at *ortho* or *para* positions was found to be more favorable for activity with best activity observed for compounds **41** and **51** with MIC values 17 μM and

16 µM respectively. The lowest activity was detected for 2-pyrrolyl substituent (**49** and **59**) with MIC values of 151 µM and 144 µM, whereas the best activity was obtained for compounds **50** and **60** containing indole ring with MIC value of 16 µM among all other bioisosteres **47–50**, **57–60**. Compounds **50** and **60** were found to have activity greater than pyrazinamide but less than ciprofloxacin and streptomycin respectively. Overall, these results indicated cyano-pyridines and cyanopyrans as promising lead molecules for the development of newer antitubercular agents. A summary of the antitubercular activities of synthesized compounds (**1–60**) is depicted in Fig 4 whereas the structure activity relationships is shown in Fig 5.

## Molecular docking studies

Understanding the mechanism of action of the antitubercular activity of the newly synthesized compounds, molecular docking studies was carried to predict the binding energy of ligands within the binding site of target proteins (Tables 4–6, refer to S1 File, Fig 6). Compound **40** had shown high binding energy (-9.67 kcal/mol) which was correlated with evaluated MIC value of 8 µM. It was found to be too close to methotrexate -10.0287 kcal/mol. Compound **26** had binding energy of -7.97 kcal/mol which corresponded to a MIC value of 33 µM whereas the binding energies for compounds **50** and **60** were found to be -9.82 and -9.49 kcal/mol. We docked the synthesized compounds in the Isoniazid (INH) active site, considering the previously documented thymidylate kinase (TK) inhibitory action of structurally similar pyridine

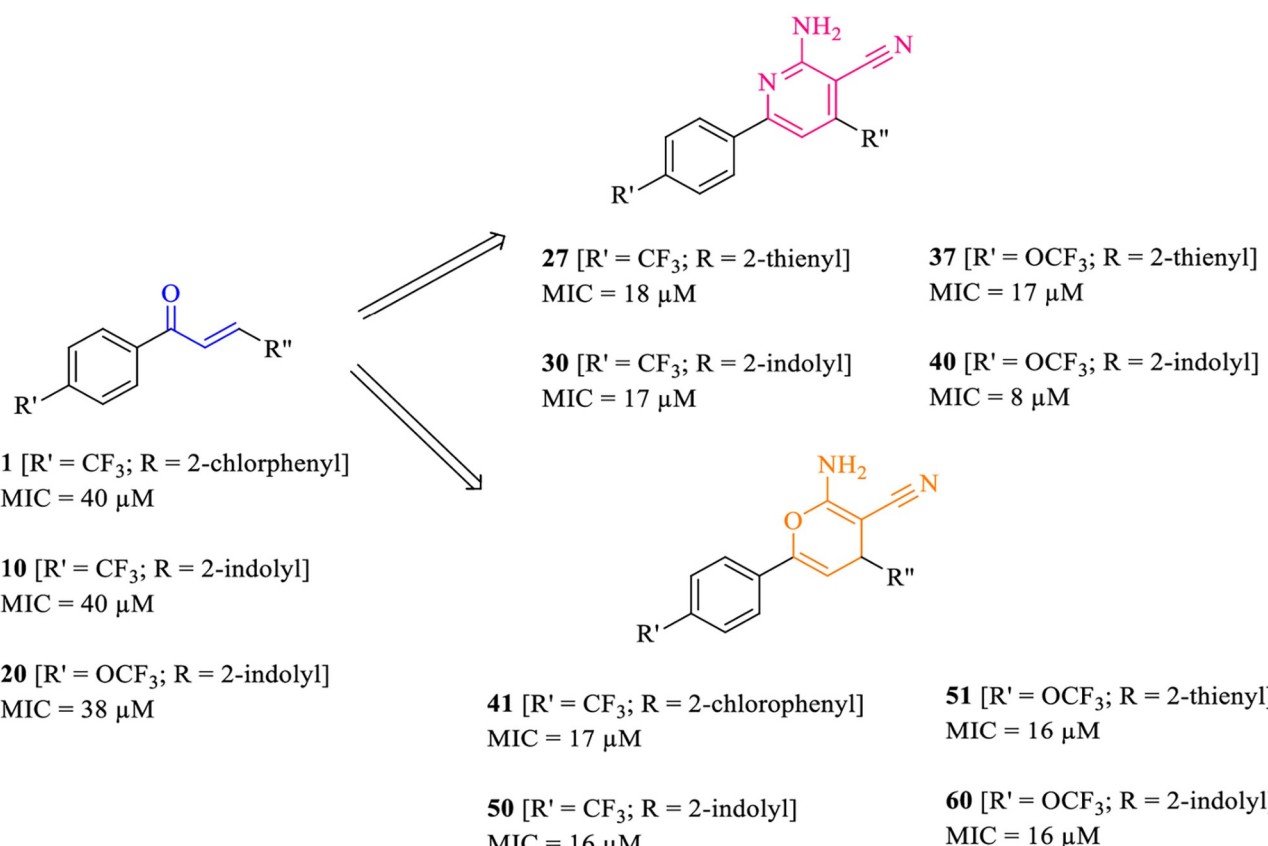

27 [R' = CF₃; R = 2-thienyl]
MIC = 18 µM

37 [R' = OCF₃; R = 2-thienyl]
MIC = 17 µM

30 [R' = CF₃; R = 2-indolyl]
MIC = 17 µM

40 [R' = OCF₃; R = 2-indolyl]
MIC = 8 µM

1 [R' = CF₃; R = 2-chlorphenyl]
MIC = 40 µM

10 [R' = CF₃; R = 2-indolyl]
MIC = 40 µM

20 [R' = OCF₃; R = 2-indolyl]
MIC = 38 µM

41 [R' = CF₃; R = 2-chlorophenyl]
MIC = 17 µM

51 [R' = OCF₃; R = 2-thienyl]
MIC = 16 µM

50 [R' = CF₃; R = 2-indolyl]
MIC = 16 µM

60 [R' = OCF₃; R = 2-indolyl]
MIC = 16 µM

**Fig 4. Summary of the antitubercular activities of fluorinated chalcones (1–20), 2-amino-pyridine-3-carbonitrile (21–40) and 2-amino-4H-pyran-3-carbonitrile (41–60) derivatives.**

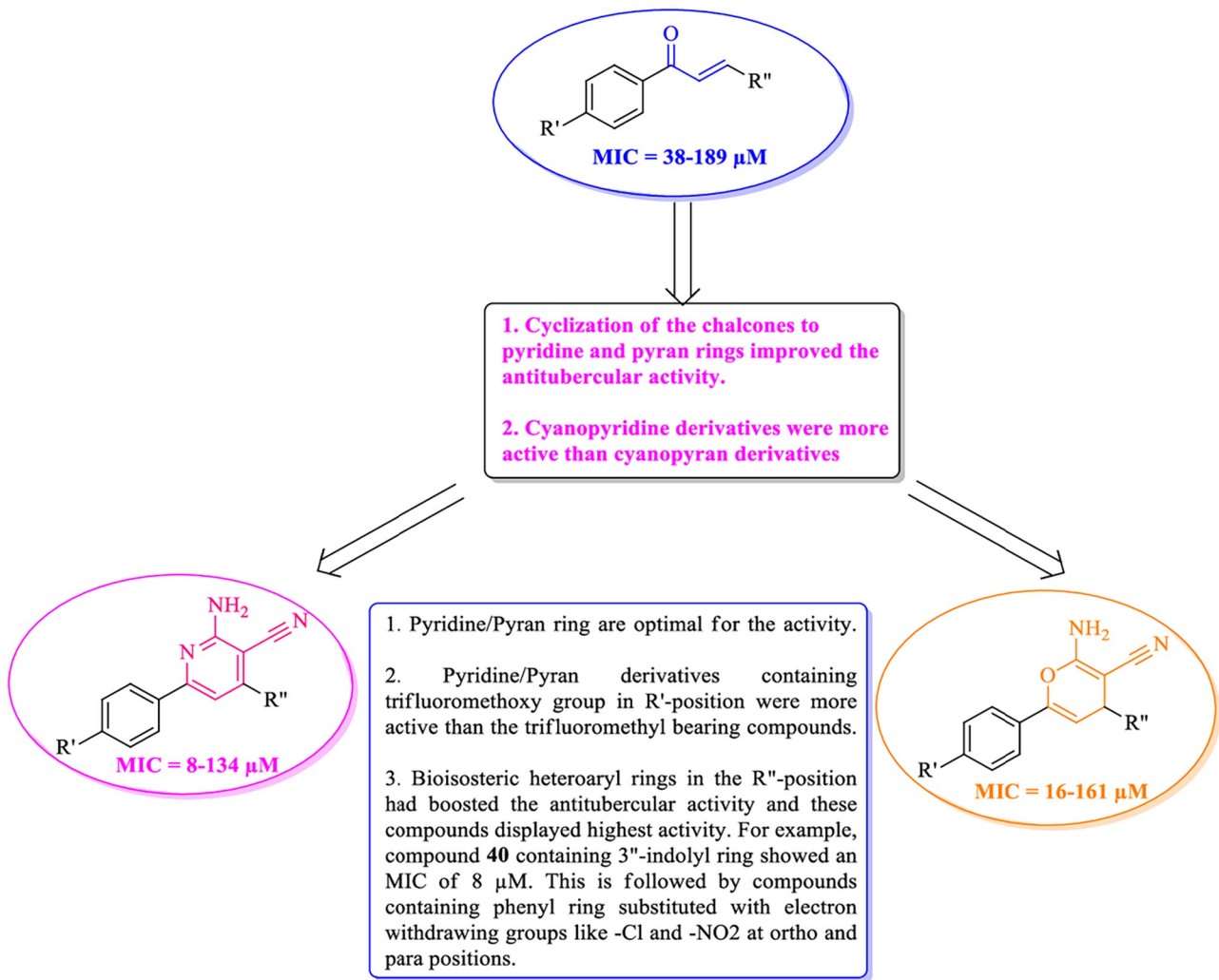

**Fig 5. Structure activity relationships of fluorinated chalcones (1–20), 2-amino-pyridine-3-carbonitrile (21–40) and 2-amino-4H-pyran-3-carbonitrile (41–60) derivatives.**

and pyran antimicrobial drugs, as well as isoniazid [50, 54]. TK protein was retrieved from the protein data bank (PDB ID: 1G3U) to validate and designate the target protein for the antibacterial activity of newly synthesized 2-aminopyridine-3-carbonitrile derivative and 2-amino-4H-pyran-3-carbonitrile derivative. The docking conformation of compound **40** suggests good interactions with the active site residues of this protein. The docking interactions of compound **40** clearly reveals the importance of 6-trifluoromethoxyphenyl-4-heteroaryl cyanopyridines in enhancing the antimycobacterial activity over 6-trifluoromethylphenyl-4-heteroaryl cyanopyridines/pyrans. This compound had shown both hydrophobic and hydrogen bonding interactions. For instance, nitrogen of the pyridine scaffold had shown hydrogen bonding with amino acid ASP9 and the amino group at the 2nd position of pyridine ring had additional hydrogen bonding with the amino acids ASP163, GLU166 and MG:300 which enhanced the binding of this compound. In addition, the indole moiety at position-4 had strong hydrophobic (pi-pi stacking) interactions with the amino acid PHE70. The halogen bonding of the trifluoromethoxy group located on the 4th position of the phenyl ring substituted at the position-

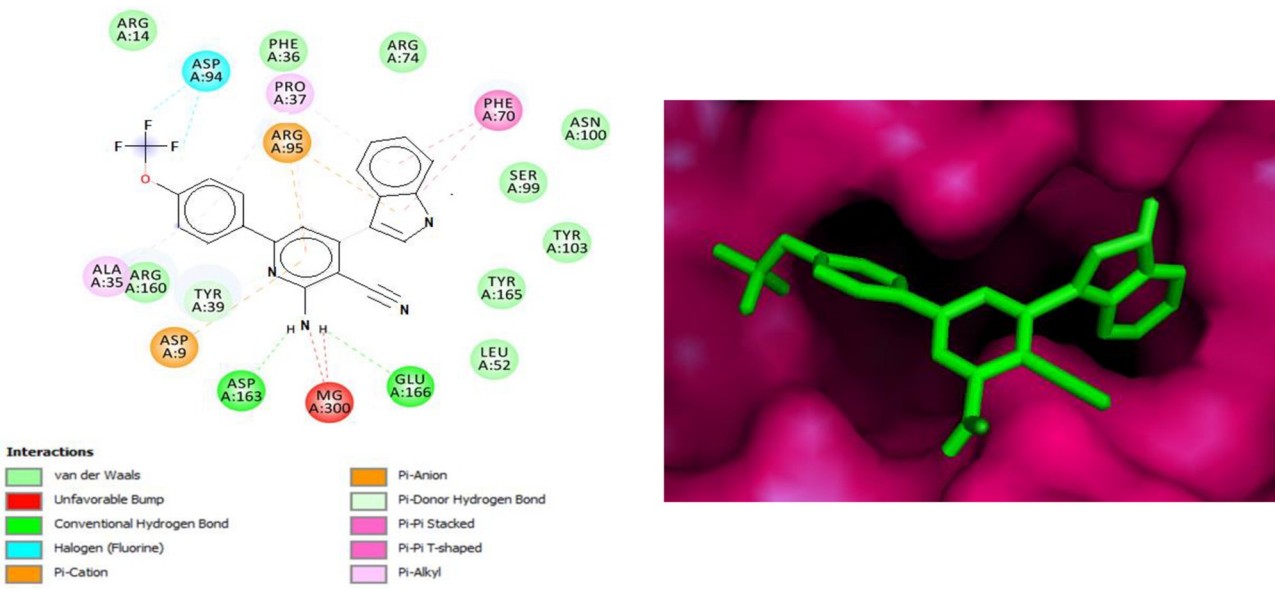

**Fig 6. 2D and 3D interactions of 40 with PZA active site of 1G3U, which shown hydrogen bond and other interactions.**

6 of pyridine motif with ASP94 had further substantiated the binding and enhanced the activity (Fig 6). The other interactions for compound **40** with the protein active site are depicted in Table 7 (refer to S1 File). Hence, the docking studies are in line with the structure activity relationships of the *in vitro* antitubercular activity data.

## Cytotoxicity assay

The compound **40** was estimated with an $IC_{50}$ value of >70 μg/mL against the tested human normal liver cell line (LO2) in its *in vitro* MTT assay (Table 8, refer to S1 File). This result suggested that compound **40** being most potent has selective activity towards the *Mycobacterium tuberculosis* H37Rv strain over the human cells. Hence it is a safe analogue to process the molecule further to develop the novel antitubercular agent.

## *In silico* drug likeliness studies

Using the web based Swiss ADME program, some parameters of compounds **20**, **37**, **40** and **60** were computed along with control standards such as *Ciprofloxacin*, *Streptomycin* and *Pyrazinamide* (Table 9, refer to S1 File). The GI absorption was low for compounds **37** and *Streptomycin*. Compounds **20**, **60**, *Ciprofloxacin* and *Streptomycin* were found to be P-Glycoprotein (PgP) substrates. No compounds and standards allowed Lipinski violations except *Streptomycin*. In case of CYP2D6 inhibition, all compounds fared well except for **20**. Compounds **37**, **40** and **60** were found to be CYP2C19 inhibitors. Among all the compounds, **40** fared better in terms of activity and SWISSADME properties compared to **20**, **37** and **60**. It was observed that **40** inhibited CYP2C19, but did not inhibit CYP2D6. However, it had a high GI absorption rate, did not function as a PgP substrate, and passed the Lipinski Rule of 5. As a result, compound **40** possesses good drug-like qualities and can be used as a new lead compound for additional *in-vivo* research.

## Conclusions

In this present study, a novel series of chalcones, pyridine-3-carbonitrile, 4H-pyran-3-carbonitrile scaffolds were evaluated for their antitubercular activity using *in-vitro* and *in-silico* methods. The antitubercular activities of compounds **27**, **30**, **40**, **41**, **50**, **51**, **60** consisting indole and 2-chlorophenyl moiety were exhibited most potency among all compounds being studied. The docking experiments suggested that these compounds might also inhibit 1G3U, considering as promising drug candidates as novel antitubercular drugs. The highest potent compound **40** was selective active against *Mycobacterium tuberculosis* H37Rv strain compared to the normal human cells and demonstrated good drug-like qualities in the *in silico* ADME experiments. Compound **40** can be further examined for *in vivo* characterization. Future research may be extended and continued in the direction of the synthesis of novel analogues and their toxicity testing.

## Supporting information

**S1 File.**
(DOCX)

## Acknowledgments

L.S.B & Y.R.P like to acknowledge University College of Pharmaceutical Sciences Andhra Pradesh, India for providing the lab facilities and chemicals for this work. R.R.B and A.B.S would like to thank the Dean's office of College of Pharmacy and Health Sciences, Ajman University, UAE & Vignan Pharmacy College, Vadlamudi, Andhra Pradesh, India for their support.

## Author Contributions

**Conceptualization:** Rajendra Prasad Yejella, Afzal Basha Shaik.

**Data curation:** Surendra Babu Lagu.

**Formal analysis:** Surendra Babu Lagu, Srinath Nissankararao, Richie R. Bhandare, Venu Sampath Golla, Bontha Venkata Subrahmanya Lokesh, Afzal Basha Shaik.

**Funding acquisition:** Surendra Babu Lagu, Afzal Basha Shaik.

**Investigation:** Surendra Babu Lagu.

**Methodology:** Surendra Babu Lagu.

**Project administration:** Rajendra Prasad Yejella.

**Resources:** Richie R. Bhandare.

**Software:** Surendra Babu Lagu, Srinath Nissankararao, Richie R. Bhandare, Bontha Venkata Subrahmanya Lokesh.

**Supervision:** Rajendra Prasad Yejella, Afzal Basha Shaik.

**Visualization:** M. Mukhlesur Rahman.

**Writing – original draft:** Surendra Babu Lagu, Rajendra Prasad Yejella, Srinath Nissankararao, Richie R. Bhandare, Venu Sampath Golla, Bontha Venkata Subrahmanya Lokesh, M. Mukhlesur Rahman, Afzal Basha Shaik.

**Writing – review & editing:** Surendra Babu Lagu, Rajendra Prasad Yejella, Srinath Nissankar-arao, Richie R. Bhandare, Venu Sampath Golla, Bontha Venkata Subrahmanya Lokesh, M. Mukhlesur Rahman, Afzal Basha Shaik.

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
