## [Decision Letter · Decision Letter 0]

2 Sep 2021

PONE-D-21-21460

Antitubercular activity assessment of fluorinated chalcones, 2‐aminopyridine‐3‐carbonitrile and 2‐amino-4H-pyran‐3‐carbonitrile derivatives: In vitro, molecular docking and in-silico drug likeliness studies

PLOS ONE

Dear Dr. Shaik,

Thank you for submitting your manuscript to PLOS ONE. After careful consideration, we feel that it has merit but does not fully meet PLOS ONE’s publication criteria as it currently stands. Therefore, we invite you to submit a revised version of the manuscript that addresses the points raised during the review process.

We look forward to receiving your revised manuscript.

Kind regards,

Hanna Landenmark

Associate Editor, PLOS ONE

on behalf of 

Pooja Chawla

Journal Requirements:

We suggest you thoroughly copyedit your manuscript for language usage, spelling, and grammar. If you do not know anyone who can help you do this, you may wish to consider employing a professional scientific editing service.

Additional Editor Comments (if provided):

Authors have done good work. Although the synthetic part has already been published, the antitubercular activity was carried out. The work could have been substantiated with enzymatic studies. But in my opinion this much work may be published.

Reviewers' comments:

Reviewer's Responses to Questions

**Comments to the Author**

1. Is the manuscript technically sound, and do the data support the conclusions?

Reviewer #1: Yes

2. Has the statistical analysis been performed appropriately and rigorously? 

Reviewer #1: N/A

3. Have the authors made all data underlying the findings in their manuscript fully available?

Reviewer #1: Yes

4. Is the manuscript presented in an intelligible fashion and written in standard English?

Reviewer #1: Yes

5. Review Comments to the Author

Reviewer #1: The authors have presented antitubercular actvities for the previously reported compounds.

1. Introduction section mentions about the WHO date. Please cite the same in references.

2. Docking images are not legible. Especially, the compouds are not legible in the docking images.

3. Did the authors perform acute cytotoxicity and in vivo analysis?

4. The authors must discuss the previously performed activity of the test compounds that encouraged the appraisal of their antitubercular activity in the present report.

5. Is the clogP evaluated experimentally or theoretically? Please mention the percentage error in either case.

6. Since molecular docking forms the only basis of supporting studies, it should be discussed in details for atleast the most active compounds and corroborated with the SAR analysis.

6. PLOS authors have the option to publish the peer review history of their article (what does this mean?). If published, this will include your full peer review and any attached files.

Reviewer #1: **Yes: **Parteek Prasher

---

## [Author Response · Author response to Decision Letter 0]

17 Sep 2021

Response to reviewer and editorial comments file has been attached as a rebuttal letter for your kind consideration.

---

## [Decision Letter · Decision Letter 1]

2 Feb 2022

PONE-D-21-21460R1Antitubercular activity assessment of fluorinated chalcones, 2‐aminopyridine‐3‐carbonitrile and 2‐amino-4H-pyran‐3‐carbonitrile derivatives: In vitro, molecular docking and in-silico drug likeliness studiesPLOS ONE

Dear Dr. Shaik,

Thank you for submitting your manuscript to PLOS ONE. After careful consideration, we feel that it has merit but does not fully meet PLOS ONE’s publication criteria as it currently stands. Therefore, we invite you to submit a revised version of the manuscript that addresses the points raised during the review process.

We look forward to receiving your revised manuscript.

Kind regards,

Mohammad Shahid, Ph.D.

Academic Editor

PLOS ONE

Reviewers' comments:

Reviewer's Responses to Questions

**Comments to the Author**

1. If the authors have adequately addressed your comments raised in a previous round of review and you feel that this manuscript is now acceptable for publication, you may indicate that here to bypass the “Comments to the Author” section, enter your conflict of interest statement in the “Confidential to Editor” section, and submit your "Accept" recommendation.

Reviewer #1: (No Response)

Reviewer #2: All comments have been addressed

2. Is the manuscript technically sound, and do the data support the conclusions?

Reviewer #1: Yes

Reviewer #2: Yes

3. Has the statistical analysis been performed appropriately and rigorously? 

Reviewer #1: Yes

Reviewer #2: Yes

4. Have the authors made all data underlying the findings in their manuscript fully available?

Reviewer #1: Yes

Reviewer #2: Yes

5. Is the manuscript presented in an intelligible fashion and written in standard English?

Reviewer #1: Yes

Reviewer #2: Yes

6. Review Comments to the Author

Reviewer #1: The author have presented the anti tubercular potency of fluorinated chalcones, 2‐ aminopyridine‐ 3‐ carbonitrile and

2‐ amino-4H-pyran‐ 3‐ carbonitrile derivatives. The molecular docking and in vitro studies support the biological activity of these molecules. The authors must address the following points to improve the manuscript1.

1. The introduction section has many claims regarding the morbidity and mortality rate of TB without a valid reference. Even if the WHO statistics are cited, URL must be provided so that the readers benefit form it.

2. If the synthesis protocol is novel, the percentage yield of the compounds must be reported.

3. Drug likeliness is calculated by an online tool. Is there an ecperimental validation to the reported clogP values?

4. Is the reported online tool for calculating clogP a valid tool for thes studies? What is the incidence of error?

5. Which docking tool is used for the studies? The docking data only shows hydrogen bonding interactions. Can the other interactions of molecules with peptide be rejected?

Reviewer #2: Journal- PLOS ONE

Manuscript ID- PONE-D-21-21460R1

Title-" Antitubercular activity assessment of fluorinated chalcones, 2‐aminopyridine‐3‐

carbonitrile and 2‐amino-4H-pyran‐3‐carbonitrile derivatives: In vitro, molecular docking and in-silico drug likeliness studies"

Authors have fulfilled all the queries/comments that was asked reviewers previously. Hence, now the manuscript is well written. I believe that it is a nice piece of work for being published in the PLOS ONE. Finally, I recommend that the paper should be accepted for the publication in the present form with some justification or corrections.

Decision- Accept

Reviewer comments

1. Mention the full name NIH and PZA in the text?

2. The author mentions in the text that, the docking conformation of compound 40 “showed” good interactions and so on. This is the modelling study, thus “suggests” is the right verb to be used.

3. In docking study authors are also required to mention in what range bond distance of hydrophobic and hydrogen interactions are determined.

4. In conclusion author mentions compound 27 for the docking study while they performed docking with 26 and 40.

5. The authors need to give some more methological data regarding their analyses as how big or small was the box that the authors used for the molecular docking? Did that cube cover the whole protein or only the binding site? These information need to be provide in methodology.

7. PLOS authors have the option to publish the peer review history of their article (what does this mean?). If published, this will include your full peer review and any attached files.

Reviewer #1: **Yes: **Parteek Prasher

Reviewer #2: **Yes: **Mantasha Idrisi,

National Center for Natural Products Research, School of Pharmacy, University of Mississippi, University, MS38677,USA

---

## [Author Response · Author response to Decision Letter 1]

10 Feb 2022

A separate document containing a point by point response is uploaded along with the manuscript

---

## [Editor Report · Decision Letter 2]

23 Feb 2022

Antitubercular activity assessment of fluorinated chalcones, 2‐aminopyridine‐3‐carbonitrile and 2‐amino-4H-pyran‐3‐carbonitrile derivatives: In vitro, molecular docking and in-silico drug likeliness studies

PONE-D-21-21460R2

Dear Dr. Shaik,

We’re pleased to inform you that your manuscript has been judged scientifically suitable for publication and will be formally accepted for publication once it meets all outstanding technical requirements.

Kind regards,

Mohammad Shahid, Ph.D.

Academic Editor

PLOS ONE
---

## [Editor Report · Acceptance letter]

8 Mar 2022

PONE-D-21-21460R2 

Antitubercular activity assessment of fluorinated chalcones, 2‐aminopyridine‐3‐carbonitrile and 2‐amino-4H-pyran‐3‐carbonitrile derivatives: *In vitro*, molecular docking and in-silico drug likeliness studies 

Dear Dr. Shaik:

I'm pleased to inform you that your manuscript has been deemed suitable for publication in PLOS ONE. Congratulations! Your manuscript is now with our production department. 

Kind regards, 

on behalf of

Dr. Mohammad Shahid 

Academic Editor

PLOS ONE